# Long-wave infrared transparent sulfur polymers enabled by symmetric thiol cross-linker

Miyeon Lee[1,2], Yuna Oh[3], Jaesang Yu[3], Se Gyu Jang[4], Hyeonuk Yeo [5], Jong-Jin Park[2] & Nam-Ho You [1] ✉

Infrared (IR) transmissive polymeric materials for optical elements require a balance between their optical properties, including refractive index ($n$) and IR transparency, and thermal properties such as glass transition temperature ($T_g$). Achieving both a high refractive index ($n$) and IR transparency in polymer materials is a very difficult challenge. In particular, there are significant complexities and considerations to obtaining organic materials that transmit in the long-wave infrared (LWIR) region, because of high optical losses due to the IR absorption of the organic molecules. Our differentiated strategy to extend the frontiers of LWIR transparency is to reduce the IR absorption of the organic moieties. The proposed approach synthesized a sulfur copolymer via the inverse vulcanization of 1,3,5-benzenetrithiol (BTT), which has a relatively simple IR absorption because of its symmetric structure, and elemental sulfur, which is mostly IR inactive. This strategy resulted in approximately 1 mm thick windows with an ultrahigh refractive index ($n_{av} > 1.9$) and high mid−wave infrared (MWIR) and LWIR transmission, without any significant decline in thermal properties. Furthermore, we demonstrated that our IR transmissive material was sufficiently competitive with widely used optical inorganic and polymeric materials.

Infrared (IR) thermal imaging is used to detect and visualize the energy (heat) naturally radiated from an object, and is widely employed in important systems to identify and analyze targets when visible light is nearly or completely absent[1]. The detectable IR wavelength band is determined by the temperature range of the target object. The mid−wave infrared (MWIR) region, which is a high temperature range (>500 K), is mainly observed for military purposes, and the long−wave infrared (LWIR) region, with infrared light corresponding to 300 K (≈10 µm), is maximally emitted and closely related to applications in normal life[2]. There has recently been growing demand for high-performance optics elements in general industry, medical care,

security, and firefighting, among other fields. LWIR transparent polymers have emerged as promising candidate materials for advanced IR optical technology[3].

Typically, inorganic semiconductors (e.g., Ge and Si) and chalcogenide glasses (ChGs) are utilized as IR transmissive materials in IR optical applications. Although these materials have an excellent refractive index ($n > 2.0$) and IR transparency, their wider application is limited by several disadvantages, including difficult processing, high manufacturing cost and toxicity[4,5]. Many traditional polymers can address these shortcomings of inorganic materials because they're relatively low cost and have excellent processability. However, their

[1]Carbon Composite Materials Research Center, Korea Institute of Science and Technology (KIST), Wanju 55324, Republic of Korea. [2]Department of Polymer Engineering, Chonnam National University, Gwangju 61186, Republic of Korea. [3]Institute of Advanced Composite Materials Research Center, Korea Institute of Science and Technology (KIST), Wanju 55324, Republic of Korea. [4]Functional Composite Materials Research Center, Korea Institute of Science and Technology (KIST), Wanju 55324, Republic of Korea. [5]Department of Chemistry Education, Kyungpook National University, Daegu 41566, Republic of Korea. ✉e-mail: polymer@kist.re.kr

optical losses, due to IR absorption of C−H or heteroatom−hydrogen covalent bonds, and their low refractive index ($n < 1.6$), usually make them limited for various IR transmissive applications[6,7].

Recently, inverse vulcanization polymers have emerged as a class of IR transmissive materials, which avoid the limitations of traditional polymers. Inverse vulcanization is one of the important methods used to prepare high sulfur content polymers, through the direct copolymerization of elemental sulfur with organic cross-linkers[8]. These sulfur-containing polymers synthesized via inverse vulcanization exhibit attractive optical properties because of the presence of dynamic S−S bonds, which differentiate them from carbon-based polymers. The high molar refraction and IR-inactive nature of the S−S bonds result in an excellent refractive index and IR transparency, respectively, making inverse vulcanized polymers suitable for IR optical elements and thermal imaging applications[9]. Although the continuing development of sulfur polymers via inverse vulcanization using various cross−linkers has resulted in improvements in thermal properties and MWIR transmittance[8,10−13], difficult issues remain in efforts to develop LWIR transparent materials. For example, most of the organic moieties in the cross−linkers absorb in the IR fingerprint (7–20 μm, 1500−500 cm$^{-1}$) region[14].

Nevertheless, some promising studies on transmissive materials for the long−infrared (LWIR) region have been reported[15−17]. Inverse vulcanized polymers for LWIR applications were investigated by reacting elemental sulfur with the cross-linker tetravinyltin (TVSn), which contained an organometallic molecule. The prepared S-TVSn copolymers showed high refractive index ($n$) and transparency from the mid-infrared to the long-infrared region, however, bubbles were unavoidable in the film due to the low $T_g$, and the rubber-like material became brittle after a few days[15]. An inverse vulcanized polymer with high $n$ and IR transparency was also been reported, and had excellent mechanical properties such as great extensibility, recovery behavior and self-healing. However, a large attenuation of LWIR light was observed with increasing film thickness[16].

Recently, computational methods have been used for the design and synthesis of LWIR transmissive materials. Based on a simulation spectrum analysis that suggested C=C bonds induce strong absorption in the LWIR region, the organic cross-linker NBD2 (norbornadiene), which has low LWIR absorbance and maintains reactivity with sulfur, was designed and synthesized. The synthesized S-NBD2 copolymers demonstrated improved LWIR transparency with some peaks of transparency in the LWIR region spectrum[17]. However, the refractive index ($n$) tended to be somewhat lower as the cross-linker content increased.

As mentioned above, previous studies have shown transparency in the MWIR and LWIR regions, respectively. However, practical applications in the optical field remain limited because of the trade-off between IR transmittance and other properties, such as refractive index ($n$) and $T_g$. Although research on the IR applications of polymers prepared by inverse vulcanization has steadily continued, effectively adjusting that trade-off for optical element applications is a significant topic, and remains a challenge to further progress.

Herein, we report on the synthesis of a high sulfur content copolymer via the inverse vulcanization of elemental sulfur with a trifunctional aromatic thiol cross-linker, 1,3,5-benzenetrithiol (BTT). The symmetry of the molecule is one of the important factors that can effectively reduce absorption in the IR spectrum. Compounds with a highly symmetric structure exhibit less complex IR spectra, due to the absence of molecular dipole moments that result from symmetry, and a relatively large number of infrared-inactive vibrations[18]. BTT was chosen as the cross-linker, with the expectation it would improve IR transparency by combining the IR absorption characteristics of the symmetric structure of the cross-linker and S−S bonds, which are mostly IR inactive. The three thiol groups in BTT react with elemental sulfur to form S−S bonds, producing highly cross-linked poly(sulfur-

random-(1,3,5-benzenetrithiol)) (poly(S-$r$-BTT)) copolymers, with improved thermomechanical properties.

The optical windows prepared from poly(S-$r$-BTT) copolymers allowed ultrahigh refractive indices ($n_{av} = 1.9$ to 2.0 from 637 nm to 1549 nm) and transparency in both the MWIR and LWIR regions. The most striking result of the poly(S-$r$-BTT) copolymers was the significantly high LWIR transmittance, an unprecedented achievement for previously reported inverse vulcanization polymers and high sulfur content polymers. The relatively high IR transmittance of the poly(S-$r$-BTT) copolymers was supported by two aspects: (1) the simple IR spectrum of the copolymer, due to the inverse vulcanization of symmetrically structured BTT and mostly IR-inactive sulfur, (2) the sulfur copolymers with the thiol cross-linker had very low organic content, such as carbon and hydrogen, that cause IR absorption, unlike previously synthesized sulfur copolymers, which used a vinyl or isopropenyl cross-linker. These results were confirmed by elemental analysis. Moreover, we demonstrated the wide range of practical applicability of our IR transmissive materials by successfully conducting high-quality near-infrared (NIR), MWIR, and LWIR imaging.

## Results
### Synthesis of poly(S-$r$-BTT) copolymers
Thiols can be used as efficient hydrogen atom donors and chain transfer agents and are highly reactive with elemental sulfur[19,20]. Accordingly, thiol compounds are commonly used for the synthesis of polymeric polysulfides. The interactions of sulfur with thiol-based compounds have been reported in previous studies. These studies synthesized polymer solutions via a condensation reaction between the thiol monomer and sulfur in a toluene/carbon disulfide mixture[20−23]. In contrast, in this study, the molten sulfur acts as a comonomer, as well as the reaction solvent itself[8].

BTT has three thiol groups with high reactivity, promoted by the conjugated nature of the benzyl ring, which is very advantageous for eliminating protons in a condensation reaction with elemental sulfur[22−24]. The loss of hydrogen as H$_2$S during inverse vulcanization via condensation reaction with elemental sulfur and BTT results in direct S−S bonding, which results in excellent optical properties due to the high sulfur content of the copolymer.

We first confirmed the reactivity between sulfur and the thiol cross-linker through an inverse vulcanization reaction using 1,3-benzenedithiol (BDT). The differential scanning calorimetry (DSC) curve for the poly(S-$r$-BDT) copolymer (70 wt% sulfur and 30 wt% BDT) showed no evidence of unreacted sulfur, however, the copolymer had a low glass transition temperature ($T_g = -12.38$ °C) and lacked shape persistence (Supplementary Fig. 9). The inverse vulcanization of sulfur and BTT formed a highly cross-linked network. (Supplementary Fig. 1). The poly(S-$r$-BTT) copolymers showed properties that were clearly different from sulfur copolymers based on 1,3-diisopropenylbenzene (DIB) or divinylbenzene (DVB) (Fig. 1).

Inverse vulcanized poly(S-$r$-BTT) copolymers were successfully prepared with a sulfur content of 80−50 wt% (see Supplementary information for details). The sulfur and BTT mixture was reacted in an oil bath at 185 °C, producing vigorous H$_2$S gas. The presence of H$_2$S gas could be observed by the bubbling of the reaction mixture (Supplementary Fig. 2). All experimental procedures were carried out in a fume hood and required special attention by the operator. After 1 h of reaction, the cured poly(S-$r$-BTT) copolymers were obtained as a bright yellow, transparent glassy material. The obtained copolymers were named S X-BTT Y according to the sulfur and BTT content, where X is the wt% of sulfur and Y is the wt% of BTT (as was the case with DIB and DVB).

Fourier transform infrared (FT-IR) spectra of the poly(S-$r$-BTT) copolymers showed the disappearance of S-H stretching (2500−2600 cm$^{-1}$ range)[23] and the introduction of an aromatic ring of BTT (1566−1386 cm$^{-1}$ range). In addition, new characteristic peaks of

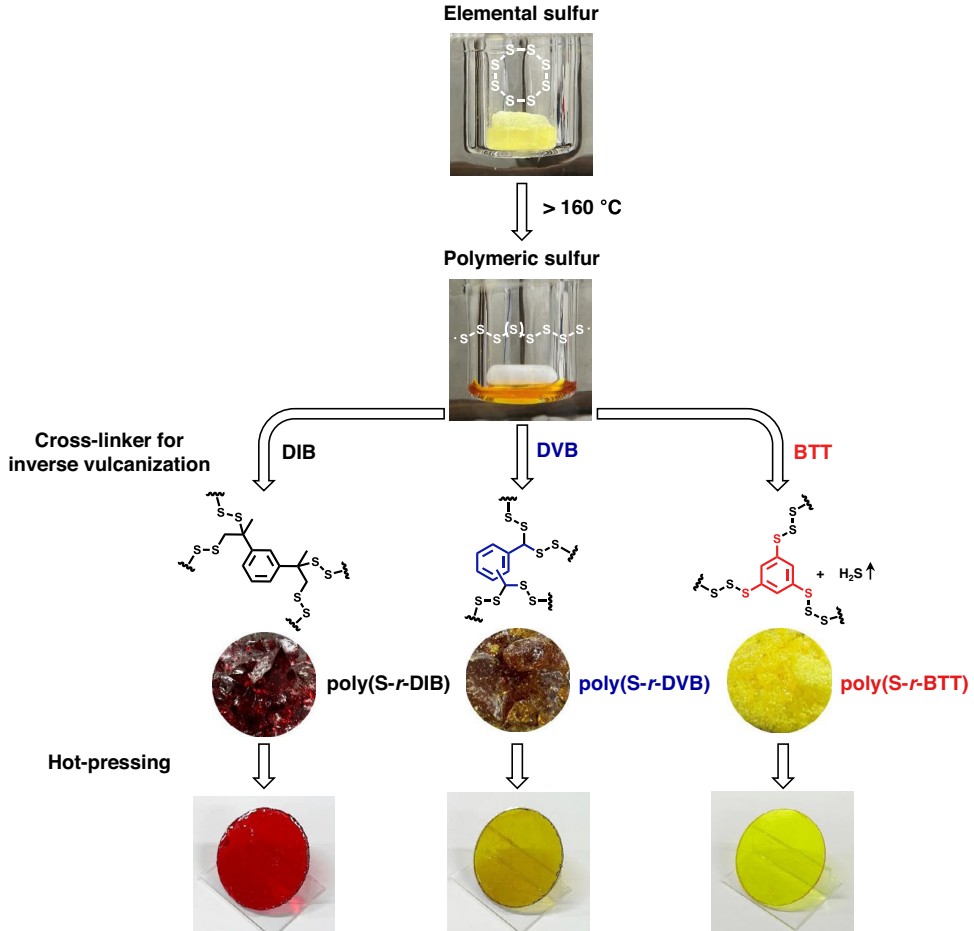

**Fig. 1 | Synthesis of sulfur polymers and preparation of polymer windows.** Poly(sulfur-random-(1,3-diisopropenylbenzene)) poly(S-*r*-DIB), poly(sulfur-random-(divinylbenzene)) poly(S-*r*-DVB) and poly(sulfur-random-(1,3,5-benzenetrithiol)) poly(S-*r*-BTT) copolymer synthesized via inverse vulcanization.

S−S bonds stretching (541−478 cm$^{-1}$ range) appeared in the poly(S-*r*-BTT) copolymers[25,26]. This suggests successful polymerization between sulfur and BTT (Fig. 2a). X-ray diffractometer (XRD) measurements of the poly(S-*r*-BTT) copolymers with various sulfur contents showed no evidence of crystalline of unreacted sulfur or BTT monomer. As with the FT-IR analysis, this indicated successful polymerization of the sulfur and BTT (Fig. 2b). Crystallization of the unreacted sulfur, which was not observed at sulfur contents of 80−50 wt%, was clearly observed in the S90-BTT10. Indeed, after the S90-BTT10 was stored for one day at room temperature it showed unreacted sulfur on the DSC and XRD curves. The unreacted sulfur was also confirmed visually (Supplementary Figs. 11 and 12).

Subsequently, various characterizations were carried out on the poly(S-*r*-BTT) copolymers with sulfur contents of 80−50 wt%.

### Thermal properties of the poly(S-*r*-BTT) copolymers

The inverse vulcanization of BTT and elemental sulfur resulted in sulfur copolymers with improved thermal properties. The thermal stability of the poly(S-*r*-BTT) copolymers was enhanced as BTT content increased, as demonstrated by Thermogravimetric analysis (TGA), which indicated an increase in char yield and decomposition temperature (Fig. 2d). All of the poly(S-*r*-BTT) copolymers were thermally stable until about 200 °C without weight loss. The S50-BTT50 with the highest BTT content showed excellent thermal stability with 5% weight loss occurring from about 369 °C.

The formation of a highly cross-linked structure was also confirmed by the solvent resistance of the poly(S-*r*-BTT) copolymers. The S70-BTT30 was completely insoluble in organic solvents, such as

dichloromethane (DCM) and tetrahydrofuran (THF) at room temperature, and maintained its original shape and weight without any decomposition reaction or swelling in THF to 50 °C for 1 h (Supplementary Fig. 16).

We also observed that the glass transition temperatures ($T_g$) of the poly(S-*r*-BTT) copolymers, determined by the 2nd DSC heating cycle, increased gradually with increasing BTT content ($T_g$ from 14.85 to 100.14 °C). As with the XRD patterns, there were no sulfur related peaks observed for sulfur contents of 80−50 wt%. The $T_g$ values of the poly(S-*r*-BTT) copolymers, defined as the tan delta of dynamic mechanical analysis (DMA) ($T_g$ from 40.83 to 118.25 °C), showed a higher value that the $T_g$ values observed with DSC. However, the $T_g$ range showed reasonably consistent trends (Fig. 2e, f) and allowed for a composition with overall good processability. The thermal properties of poly(S-*r*-BTT) copolymers are summarized in Supplementary Table 4. In the MD simulations, the $T_g$ values of the S-BTT models based on the various contents of sulfur and BTT were 46.15, 54.55, and 95.15 °C for the S70-BTT30, S60-BTT40, and S50-BTT50 models, respectively (Supplementary Fig. 17). The $T_g$ values predicted by the MD simulations were matched well with that of poly(S-*r*-BTT) copolymers observed with DSC (Fig. 2g). The $T_g$ of the S70-BTT30 model was lower than those of other models due to the flexible polymer chain induced by the high sulfur content. The $T_g$ of the S-BTT model gradually increased with increasing BTT content. This result was related to the density of the S-BTT model. At the same temperature, the total volume of the S50-BTT50 model decreased compared to the S70-BTT30 model (Supplementary Fig. 17). This result indicates that the polymer chains in the S50-BTT50 model were densely packed in the

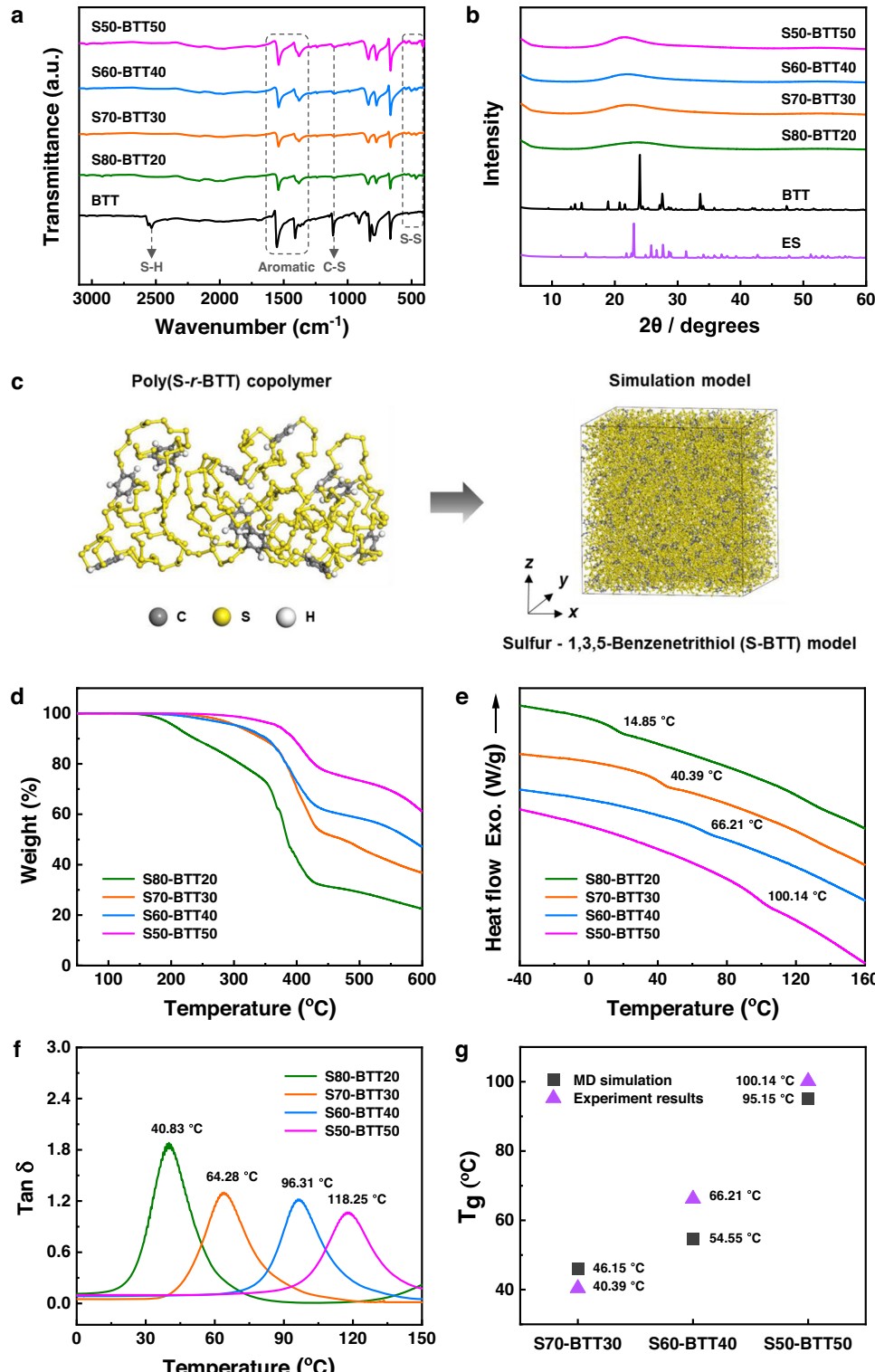

**Fig. 2 | Chemical and thermal properties of poly(S-*r*-BTT) copolymers. a** ATR-FT-IR spectra of 1,3,5-benzenetrithiol (BTT) and poly(S-*r*-BTT) copolymers. **b** XRD spectra of elemental sulfur (ES), BTT and poly(S-*r*-BTT) copolymers. **c** The poly(S-*r*-BTT) copolymer and the Sulfur-BTT simulation model. **d** TGA curves. **e** DSC curves indicating the glass transition temperatures ($T_g$). **f** Plot of tan δ indicating the $T_g$. **g** Comparison of the MD simulation and experimental results (determined by DSC curves) of the $T_g$.

simulation cell. The increase in density leads to an increase in the rigidity of the polymer chains. Therefore, the S50-BTT50 model had the highest $T_g$ due to the increase in the rigidity of the flexible polymer chains, induced by the high BTT content. The high $T_g$ of the S50-BTT50 model leads to the improvement in thermal stability.

## Optical properties of the poly(S-*r*-BTT) copolymers

Previous researchers have advised that for more meaningful IR imaging, windows at least 1 mm thick should be used, since even conventional vinyl cross-linker and aromatic ring-based polymers exhibit some IR transparency[9,17].

Bulk poly(S-*r*-BTT) copolymers were readily prepared into windows with a diameter of 25 mm and a thickness of about 1 mm by hot-pressing at 185 °C and 20 MPa (see Supplementary information for details). Despite the highly cross-linked structure of the poly(S-*r*-BTT) copolymers with high thermal properties, the presence of large quantities of dynamic S–S bonds allowed their easy and fast processing.

The processed S-BTT windows were sufficiently transparent without any bubbles or pores (Supplementary Figs. 3 and 4). The interior of this transparent yellow window was characterized by X-ray microscopy (XRM). The 3D image of S70-BTT30 window showed a clean interior without pores or defects (Supplementary Fig. 5). We also prepared about 1 mm thick windows of widely used optical polymeric materials (e.g., polyethylene (PE), cyclic olefin polymer (COP) and poly(methyl methacrylate) (PMMA)), as well as inverse vulcanization polymers (e.g., S70-DIB30 and S70-DVB30) for a relative evaluation of IR transmission (%) (Supplementary Figs. 6 and 7). A detailed description of the windows' fabrication can be found in the Supplementary Information. The average thickness of the polymer windows is given in Supplementary Fig. 8 and Supplementary Table 1.

Another advantage of the high sulfur content in the poly(S-*r*-BTT) copolymers was revealed in the optical properties of the S-BTT windows. One of the optical properties of sulfur polymers synthesized via the inverse vulcanization is their high refractive index (RI) due to the high sulfur content of these materials[7,27]. The S-BTT windows exhibited an ultrahigh RI with overall sulfur content (80–50 wt%). The RI values of the S-BTT windows are summarized in Supplementary Tables 7 and 8. After inverse vulcanization, the alkyl chain moieties remained in the S70-DIB30 and S70-DVB30 copolymers. In contrast, the S-BTT copolymers consist of an aromatic ring and S–S bonds without alkyl chain moieties. Because of these structural features, the RI of the S-BTT windows dramatically improved compared to the S70-DIB30 and S70-DVB30 windows (Fig. 3a). These results were supported by the high sulfur content of the S-BTT, as determined by the elemental analysis described below. The RI values of the S-BTT windows increased with increasing sulfur content. The highest RI, ranging from $n = 2.00$ to 1.94 in the range of 637–1549 nm was observed for the S80-BTT20 window with the highest sulfur content (80 wt%). Moreover, S50-BTT50 window with the lowest sulfur content (50 wt%) also retained a high RI ranging from $n = 1.95$ to 1.89 in the same wavelength range.

To understand this high RI values, we determined the optimized structures for the two types of model compounds based on a linear sulfur oligomeric structure (Ph-$S_{16}$-Ph, M1, S content: 76.89 wt%) and a cross-linked structure ((PhS$_6$)$_3$Ph, M2, S content: 65.32 wt%) using density-functional theory (DFT) calculations (Supplementary Fig. 19), and calculated the theoretical RI according to the Lorentz–Lorenz equation[28]. The calculation results indicated a higher RI value in M2 of the cross-linked structure, despite the low sulfur content. Basically, the high sulfur content created high molar reflection, but the results suggest that higher RI value was obtained when cross-linking occurred between the aromatic structures. For this reason, the BTT vulcanization system showed a higher RI value than the DIB system.

In addition, it was confirmed that the calculated RI values from M1 and M2, which possessed structures similar to S70-BTT30 in sulfur content, agreed relatively well with the experimental values obtained (Fig. 3b and Supplementary Table 3).

The FT-IR spectra of the polymer windows were measured five times to ensure the accuracy and reliability of the experimental results. No significant difference was observed between the five measurement results, which shows that all of the polymer windows were homogeneously fabricated through hot-pressing (Supplementary Figs. 25–30). The MWIR region was applied in the range of 3–5 μm (3300–2000 cm⁻¹) and the LWIR region was applied in the range of 7–14 μm (1420–710 cm⁻¹). The S-BTT windows exhibits numerous transparency peaks in the both MWIR and LWIR region for the overall

sulfur content of 80–50 wt% (Fig. 3c). The S80-BTT20 window with the highest sulfur content showed the highest IR transmission (%) in both the MWIR and LWIR region. As the sulfur content decreased, the IR transmittance of the S-BTT windows decreased. However, the IR transmittance of the S50-BTT50 window with the lowest sulfur content was significantly higher than the S70-DIB30 and S70-DVB30 windows (Fig. 3e, f).

This trend was also confirmed in the DFT calculations using the M1 and M2 model compounds. In both cases, absorption peaks derived from organic components other than C–S and S–S were expected in the IR region, and the molar extinction coefficient was very small, with a maximum of 300 M/cm⁻¹ compared to general organic materials (generally $10^3$ or more, reaching $10^4$ M/cm⁻¹). In particular, the peak position of the calculation agreed very well with the experimental absorption of the S-BTT series, which had a conjugated M1 and M2 structure. This indicated that the S-BTT structure was as expected, and at the same time had very low IR absorption due to the structural effect (Fig. 3d and Supplementary Figs. 20 and 21). Accordingly, the FT-IR spectra of the approximately 1 mm thick S-BTT windows showed numerous windows of transparency over the entire IR region. In contrast to the S70-DIB30 and S70-DVB30 windows, which exhibited broad absorption in the 3.3–3.5 μm region, corresponding to C–H bond stretching, the S70-BTT30 window showed over 60 % transmission windows in the MWIR region (Fig. 3e).

Surprisingly, the S70-BTT30 window had numerous transmission windows of over 60 %, even in the LWIR region. On the other hand, the LWIR transmission (%) of the previously studied S70-DIB30 and S70-DVB30 windows of the same thickness approached zero, which means theses windows showed virtually no LWIR transparency (Fig. 3f). The fingerprint region of the S70-BTT30 copolymer, which is simpler than those of the S70-DIB30 and S70-DVB30 copolymers, contributed to the high IR transmittance (Supplementary Fig. 31).

Wholly aliphatic organic polymers such as PE or COP were used as thin LWIR-barrier coatings due to the simple fingerprint region[29,30]. We measured the IR transmission (%) of 1 mm thick transparent polymer PMMA, PE and COP windows. Interestingly, our experimental results showed that the S-BTT windows had excellent IR transmittance, better than the organic polymers based on aliphatic units.

The S-BTT windows exhibited ultrahigh RI and excellent IR transmittance compared to other widely used optical polymeric materials (e.g., PE, COP and PMMA) and previously studied inverse vulcanization polymers (Supplementary Figs. 32 and 33). To the best of our knowledge, our study is the first example of an IR transmissive polymeric material with excellent transmittance in both the MWIR and LWIR regions (Supplementary Fig. 34). The outstanding IR transmittance quality of the S-BTT windows was not affected by processing conditions such as temperature or time (Supplementary Figs. 35–38). Additionally, even after being stored at room temperature for one month, the windows maintained excellent IR transmittance in both the MWIR and LWIR regions without significant degradation (Supplementary Figs. 39–42).

We have explained that the structure of the poly(S-*r*-BTT) copolymers consisted only of an aromatic ring and S–S bonds. These structural features of the poly(S-*r*-BTT) copolymers were positive for IR transmittance as well as RI values. Elemental analyses of the sulfur copolymers were then carried out. The results showed the poly(S-*r*-BTT) copolymers had relatively high sulfur (S) content and low carbon (C) and hydrogen (H) content. Furthermore, the S50-BTT50 with the highest BTT feed ratio showed lower carbon content and higher sulfur content than S70-DIB30 and S70-DVB30 (Supplementary Fig. 43 and Supplementary Table 10). These results further indicate the structural features of the poly(S-*r*-BTT) copolymers are composed of numerous S–S chains centered on an aromatic ring.

The high S/C atomic ratio of the poly(S-*r*-BTT) copolymers, calculated from the elemental analysis results, contributed to the

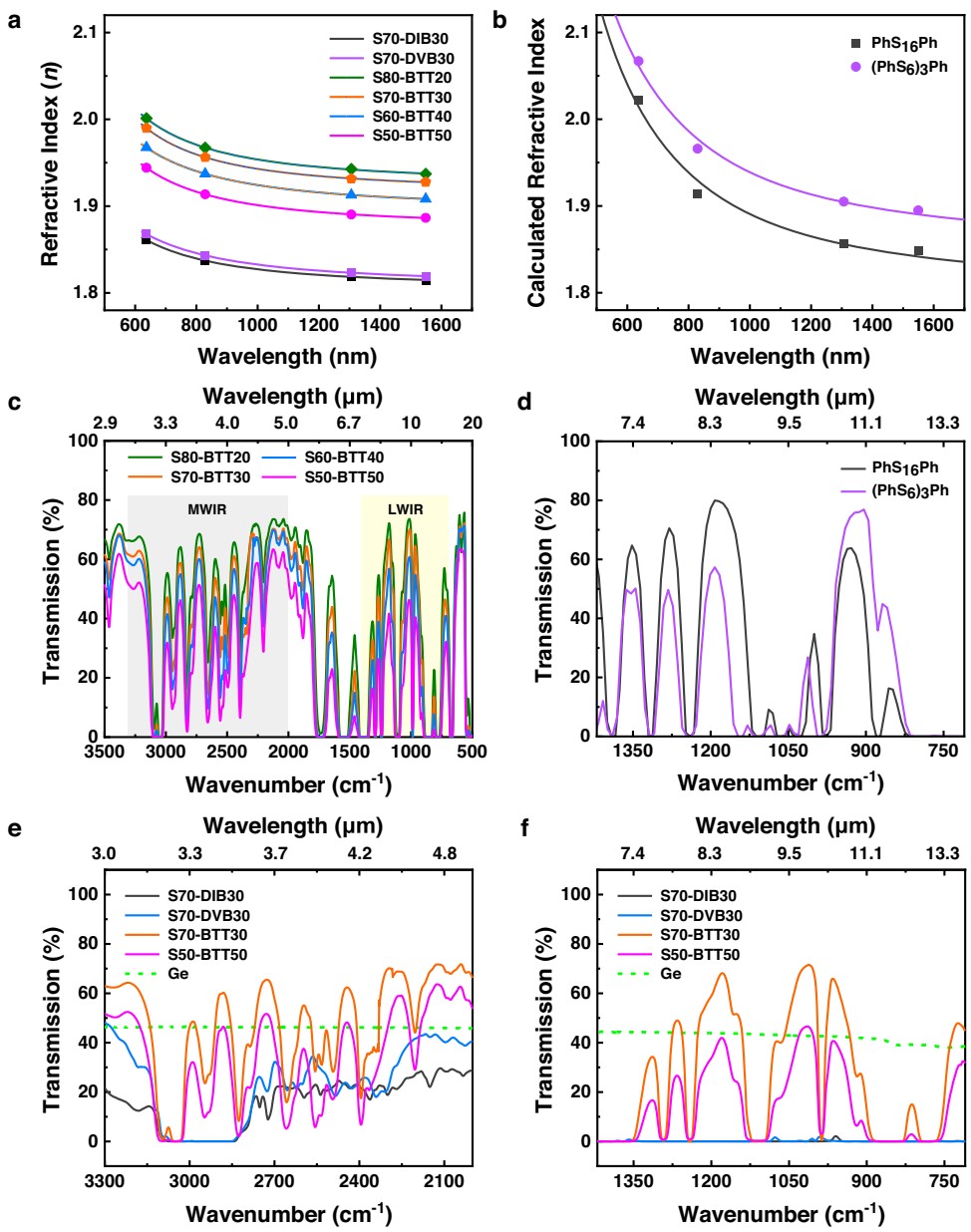

**Fig. 3 | Optical properties of Ge and polymer windows. a** Average refractive index curves of S70-DIB30, S70-DVB30 and poly(S-*r*-BTT) windows. **b** Calculated refractive index curves. **c** Average IR transmission (%) spectra of poly(S-*r*-BTT) windows (thickness ca. 1 mm). **d** Calculated IR transmission (%) spectra. **e** MWIR transmission spectra of Ge, S70-DIB30, S70-DVB30, S70-BTT30 and S50-BTT50 windows (thickness ca. 1 mm). **f** LWIR transmission spectra of Ge, S70-DIB30, S70-DVB30, S70-BTT30 and S50-BTT50 windows (thickness ca. 1 mm).

enhanced optical properties, particularly due to the specific properties of the S–S bonds (e.g., mostly IR inactive and high assigned *n*) (Supplementary Figs. 44–46 and Supplementary Table 11). These findings confirmed the S90-BTT10 window exhibited unprecedentedly high IR transmittance in both the MWIR and LWIR region (Supplementary Fig. 13), but an IR-imaging experiment was not considered because of the presence of unreacted sulfur.

**IR-imaging experiments**

We carried out IR-imaging experiments to demonstrate that, e.g., poly(S-*r*-BTT) copolymers are sufficiently transparent for IR applications. For comparison, IR visualization was also attempted through the PMMA window and the previously studied S70-DIB30 and S70-DVB30 windows. MWIR and LWIR images were taken through the 1 mm thick polymer windows previously described (Fig. 4e) and used for the IR

transmittance measurements, and the IR-imaging setup is shown in Fig. 4a, b.

For the MWIR imaging experiments of the USAF 1951 target, a thermal emission microscope with a transmission range of 3–5 μm was used. The transparent plastic PMMA failed to visualize the USAF target due to its low transmittance, while the S70-DIB30 and S70-DVB30 windows were capable of imaging in the MWIR region, with results consistent with previous studies. A decrease in the transmittance was observed as the sulfur contents of the poly(S-*r*-BTT) copolymers decreased, however, high-quality MWIR imaging was achieved for the overall sulfur content of 80–50 wt%. The S80-BTT20 window with the highest transmission (%) in MWIR region provided the clearest image, and the S50-BTT50 window with the lowest sulfur content (50 wt%) also provided a clear image (Fig. 4f). The obtained MWIR images were reasonably consistent with FT-IR analysis trend.

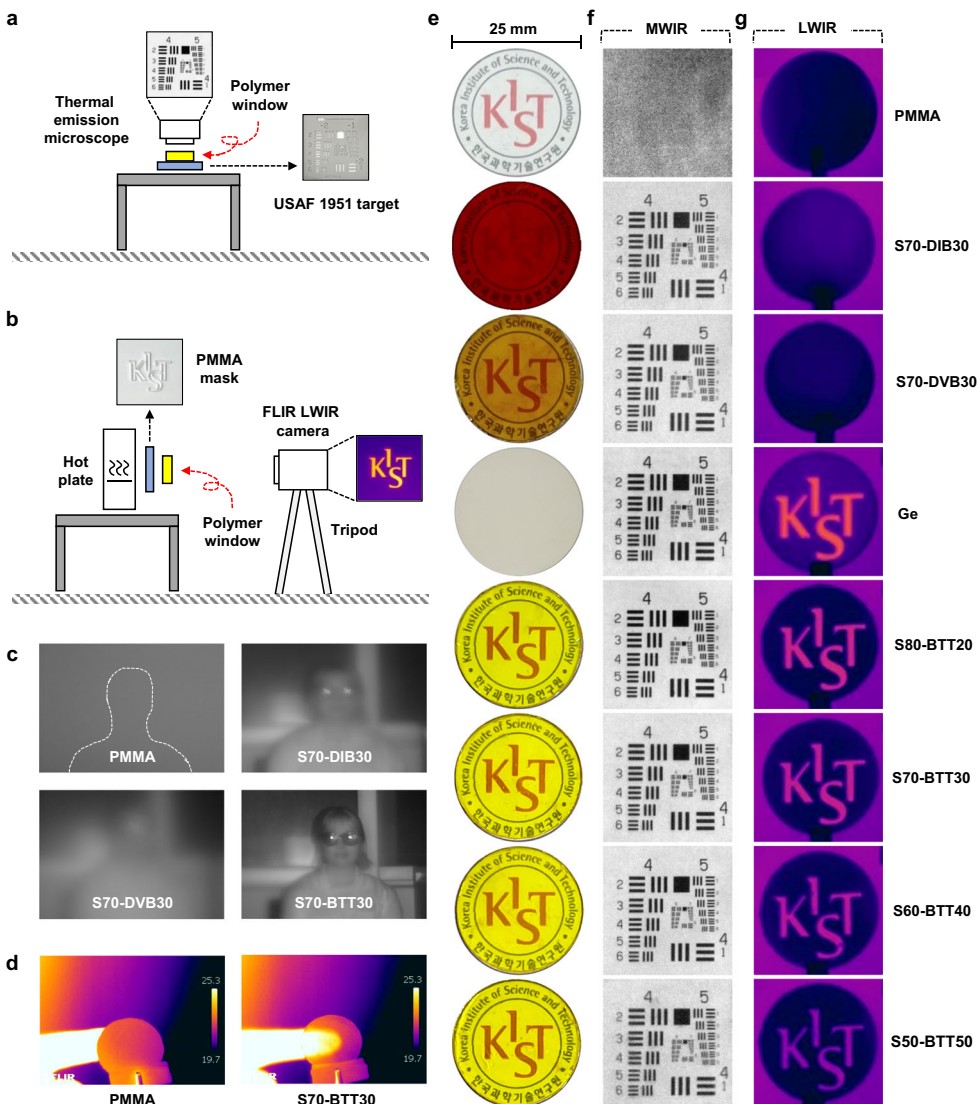

**Fig. 4 | IR-imaging experiments. a** Schematic illustration of the MWIR imaging setup. **b** Schematic illustration of the LWIR imaging setup. **c** NIR images of a female subject captured through the polymer windows; PMMA, S70-DIB30, S70-DVB30 and S70-BTT30 (thickness ca. 1 mm). **d** LWIR images of a human finger captured through the PMMA and S70-BTT30 windows (thickness ca. 1 mm). **e** Photographs of various polymer and Ge windows (diameter 25 mm × thickness ca. 1 mm). **f** MWIR images of USAF 1951 target through various polymer and Ge windows (thickness ca. 1 mm). **g** LWIR images of patterned PMMA mask through various polymer and Ge windows (thickness ca. 1 mm); PMMA, S70-DIB30, S70-DVB30, Ge, S80-BTT20, S70-BTT30, S60-BTT40 and S50-BTT50 (from top to bottom).

For LWIR imaging experiments with various patterned PMMA masks[17], an FLIR camera with the transmission range of 7.5–13 μm was used. The S70-DIB30, S70-DVB30 and PMMA windows, which were capable of imaging in the MWIR region, clearly showed opaque properties in the LWIR region, while the S-BTT windows showed excellent transparency in the LWIR region (Supplementary Movie 1). As with the MWIR imaging experiments, a decrease in transmittance of LWIR was observed as the sulfur contents of the poly(S-*r*-BTT) copolymers decreased, however, high-quality LWIR imaging was achieved for overall sulfur contents of 80–50 wt% (Fig. 4g). The S80-BTT20 window with the highest transmission (%) in LWIR region provided the clearest image. The obtained LWIR images were of a comparable quality to those obtained through a Ge window with an expensive inorganic transmissive material. Moreover, the S70-BTT30 window successfully captured a human finger (Fig. 4d). The S50-BTT50 window with the lowest sulfur content (50 wt%) was also sufficiently transparent in the LWIR region. The obtained LWIR images were reasonably consistent with the FT-IR analysis trend. In an additional near-IR-imaging experiment, the S70-BTT30 window successfully captured female subjects (Fig. 4c).

## Discussion

In conclusion, we report the synthesis of LWIR transmissive sulfur copolymers via the inverse vulcanization of symmetric trifunctional thiol, 1,3,5-benzenetrithiol (BTT) with elemental sulfur. Poly(S-*r*-BTT) copolymers with high thermal stability and high glass transition temperature ($T_g$) were prepared into optical windows by hot-pressing. The poly(S-*r*-BTT) windows exhibited an ultrahigh refractive index ($n_{av} > 1.9$) that improved transmission in both the MWIR and LWIR regions, as demonstrated by FT-IR spectra and IR-imaging experiments. The significantly high LWIR transmittance of the poly(S-*r*-BTT) windows is a particularly noteworthy achievement. Inverse vulcanization using a symmetric thiol cross-linker effectively reduced the IR absorption of the copolymer. In addition, the high sulfur content of the poly(S-*r*-BTT) copolymers contributed to the enhanced optical properties, due to the specific properties of the S−S bonds (mostly IR

inactive and high assigned $n$). This development provides a clearer and more reliable IR transparent material for IR applications using at least 1 mm thick windows.

## Methods

### Materials

Sulfur ($S_8$, powder, 98.0 % extra pure, SAMCHUN), 1,3,5-Benzene-trithiol (BTT, >98.0 % (GC), TCI), 1,3-Benzenedithiol (BDT, >95.0 % (GC), TCI), 1,3-Diisopropenylbenzene (DIB, >97.0 % (GC), TCI), Divi-nylbenzene (DVB, 80.0 % technical grade, Aldrich), Polyethylene (PE, low density, Aldrich), Poly(methyl methacrylate) (PMMA, Aldrich), Cyclic olefin polymer (ARTON D4000, JSR Corporation), Polyimide film (Kapton), Dichloromethane (DCM) and Tetrahydrofuran (THF) were purchased and used as received. Uncoated Germanium windows (Ge, Dia. 25.00 (mm) × Thickness 1.00 (mm)) and USAF 1951 target were purchased from Edmund Optics. Various patterned PMMA masks were prepared as target for LWIR imaging[17].

### Synthesis procedure

The following is the general procedure for the inverse vulcanization of thiol cross-linker, 1,3,5-benzenetrithiol (BTT) with elemental sulfur. Specific details are given in the supporting information experimental section. Sulfur ($S_8$) was added to a glass vial equipped with a magnetic stir bar and then heated in a 185 °C oil bath until the yellow sulfur powder turn into an orange liquid sulfur. 1,3,5-Benzenetrithiol (BTT) was added directly to the liquid sulfur. The sulfur and BTT mixture reacted in a 185 °C oil bath, generating vigorous gas. After 1 h, the glass vial with product was quenched in a liquid nitrogen bath and the product was separated from the vial. All experimental procedures were carried out in fume hoods. It should be noted that toxic $H_2S$ gas is generated during inverse vulcanization, requiring operator care for safety.

### Characterization

Differential scanning calorimetry (DSC, DSC 8000, PerkinElmer, USA) was carried out under $N_2$ flow at a heating (cooling) rate of 10 °C/min from −45 to 165 °C. Thermogravimetric analysis (TGA, Setaram Solutions, France) was performed under $N_2$ flow at a heating rate of 10 °C/min from 30 to 600 °C. Dynamic mechanical analysis (DMA, DMA Q800, Waters, USA) was carried out −45 °C to 165 °C at a scanning rate of 5 °C/min with a preload of 0.01 N. The specimens were prepared as thin films (30 mm length, 10 mm wide, and ca. 0.4 mm thickness). Solid-state $^{13}C$ NMR CP/MAS spectra were measured using solid 500 MHz NMR (Bruker Avance III HD, Bruker, German) with 4 mm CPMAS probes. Spinning at 5 kHz and pulse repetition delays of 5 s were performed. Elemental analysis (EA, IT/Flash 2000, Thermo Fisher Scientific, USA) was conducted to determine the N, C, H, S content of the copolymers synthesized by inverse vulcanization. X-ray dif-fractometry (XRD, SmartLab, Rigaku Corporation, Japan) was per-formed in the 2Theta/Theta mode from 5° to 80° (200 mA, 45 kV). Attenuated total reflectance−Fourier transform infrared (ATR-FT-IR) spectra were obtained with a Nicolet iS10 (Thermo Fisher Scientific, USA) with 32 scans per spectrum from 4000−550 cm$^{-1}$. Transmittance−Fourier transform infrared (Transmittance-FT-IR) spectra were recorded with a Spectrum 100 (PerkinElmer, USA) with 16 scans per spectrum from 4000−400 cm$^{-1}$, or Nicolet iS10 (Thermo Fisher Scientific, USA) with 72 scans per spectrum from 4000−500 cm$^{-1}$.

The refractive indices of the polymer windows were measured using a prism coupler (PC-2000, Metricon Corporation, USA). The In-plane/out-of-plane birefringence were calculated as $\Delta n_{av} = n_{TE} - n_{TM}$ and the average refractive index ($n_{av}$) was calculated using the fol-lowing equation. $n_{av} = [(2n_{TE}^2 + n_{TM}^2)/3]^{1/2}$. The Abbe's number was given by as $V_{NIR} = (n_{829} - 1)/(n_{637} - n_{1306})$. X-ray microscopy (XRM,

Zeiss Xradia 520 Versa, Germany) with the parameters of 40.21 kV voltage and 3.03 W power were used to image the internal features of the polymer window. IR-imaging experiments were performed using various objects (e.g., female subject, human finger, USAF 1951 target and various pattered PMMA mask). Near-infrared (NIR) imaging was performed using a Digital Night Vision Binocular (APL-NV001+, Apexel Technology, China) in IR illumination of 850 nm. Mid-wave infrared (MWIR) images were captured using a Themos mini(C-10614-02) thermal emission microscope (Hamamatsu Photonics, Japan) with a 3−5 μm lens. Long-wave infrared (LWIR) images were taken using a FLIR T335 (Teledyne FLIR, USA) with a 7.5−13 μm wavelength range.

### Simulation methods

The molecular dynamics (MD) simulations were performed to pre-dict the glass transition temperatures of the poly(S-$r$-BTT) copoly-mers according to the various contents of sulfur and BTT using the Materials studio 2017 software. The S-$r$-BTT polymer chains were set with contents of carbon, hydrogen, and sulfur atoms based on the elemental analysis. The size of the simulation models was 80 Å(x) × 80 Å(y) × 80 Å(z) in the periodic boundary condition. The number of total atoms in all of the simulation models of the poly(S-$r$-BTT) copolymers was about 25,000. All models were optimized by minimization of the total potential energy. The optimized models were equilibrated using an isothermal-isobaric ensemble (NPT) at 298 K for 1 ns. The equilibrated models were heated at temperatures from 198 to 473 K at increments of 25 K for 6 ns. After the heating process, the volumes of the poly(S-$r$-BTT) copolymers models with respect to temperatures were obtained to predict the glass transition temperature. All of the MD simulations were performed using the Condensed-phase Optimized Molecular Potentials for Atomistic Simulation Studies (COMPASS) force field, which has been widely used to predict the material property of a polymer containing the sulfur atom[31,32].

Prediction of IR spectra and refractive indices by DFT calculation. We used two established kinds of model compounds, M1 and M2, which are linear and cross-linked structures, and their optimized structures at ground states by calculating at the B3LYP level with a 6-31G basis set using the software package Gaussian 09D[33]. With the optimized structures, frequency calculations were carried out at the same level, and the obtained molar absorption coefficient using the frequency in the IR area was converted to the IR absorption spectra with peak half-width at half height 4 cm$^{-1}$. Then, it was possible to calculate the target IR spectra by converting them to transmittance. In addition, the refractive indices were calculated based on the following Lorentz−Lorenz's equation[28].

$$\frac{(n_\lambda^2 - 1)}{(n_\lambda^2 + 2)} = \frac{4\pi}{3} \times \frac{\alpha_\lambda}{V_{mol}}$$

where $n_\lambda$ is the refractive index, $a_\lambda$ is polarizability at the specific wavelength ($\lambda$), and $V_{mol}$ is molecular volume, respectively. Each $V_{mol}$, and $a_\lambda$ at the specific wavelength of the models were obtained from the previous optimization and frequency calculation results.

## Data availability

All relevant data that support the findings of this study are available from the corresponding author upon request.

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

## Acknowledgements

The work was supported by a grant from the Korea Institute of Science and Technology Institutional (KIST) Open Research Program (2E32632), and this research was also supported by the Korea Evaluation Institute of Industrial Technology (KEIT) and the Ministry of Trade, Industry & Energy (MOTIE) of the Republic of Korea (No. 20011153).

## Author contributions

N.-H.Y and M.L. conceived and designed the experiments. M.L. carried out materials synthesis, characterizations and IR-imaging experiments. Y.O. and J.Y. performed MD simulations to predict glass transition temperatures. H.Y. performed DFT calculations to predict optical properties. M.L. and N.-H.Y. wrote the manuscript. J.-J.P and S.J. commented on the manuscript. N.-H.Y. supervised the project.

## Competing interests

The authors declare no competing interests.
