## [Peer Review File · Nature Communications]

Long-wave infrared transparent sulfur polymers enabled by symmetric thiol cross-linkerReviewers' Comments:

Reviewer #1:

Remarks to the Author:

The authors describe polymers obtained from bulk polymerization of elemental sulfur and 1,3,5-benzenetrithiol and demonstrate that the materials have enhanced optical properties compared to previously reported sulfur-rich polymers obtained from inverse vulcanization. The synthesis described in the manuscript is simple and straightforward, and the resulting sulfur-rich polymers exhibit a combination of properties ($RI > 1.9$, $T_g \sim 100$ C, and some windows of transparency in MWIR and LWIR regions) that represent progress compared to IR-transparent polymeric materials reported previously.

Given that inverse vulcanization polymers such as S-DVB and S-NBD2 (both described in the references cited by the authors in the present manuscript) also exhibit high T_g , acceptable transparency in the MWIR and LWIR regions, and can be prepared without significant H₂S formation (the formation of H₂S during the synthesis of S-BTT reported in the present manuscript is mechanistically necessary and at least 3 eq of H₂S is released with respect to BTT), I am doubtful as to whether the S-BTT polymers represent significant advancement in the area. Formation of H₂S represents a formidable barrier towards scale-up and application of the materials for actual optical applications, and I am not sure if such risk is outweighed by an improvement of about 1.0 in the RI value.

One aspect of the current manuscript which would be of significant contribution to the field is in the comonomer design principle. The authors claim that the use of a symmetric comonomer improves the optical transparency of polymers in the IR region. The manuscript, however, does not contain control studies to show that this design principle is what causes the improved optical properties compared to S-DIB and S-DVB (which are also both obtained from symmetric comonomers). Also, many highly crosslinked polymers, including S-DVB, show additional IR absorptions attributed to overtones and/or structural confinement. The current manuscript, although hinting at the relationship between comonomer symmetry and the IR transparency of the polymer products, do not seem to provide much experimental and/or computational evidence to show that this indeed is a design principle that other researchers in the area can follow up on.

Some additional comments are listed below.

1. Row 59-60 in the introduction reads "...their low refractive index usually make them unsuitable for IR transmissive applications.": I am not sure why $RI < 1.6$ makes a material unsuitable for IR applications as long as they show optical transparency in the IR region. Such medium- to low-index materials are excellent candidates for antireflective coatings for current inorganic IR lenses since the high intrinsic RI of inorganic materials are associated with their low transparency.
2. Row 89-90 in the introduction: More quantitative comparison between the previously reported S-NBD2 and S-BTT in the current manuscript would be desirable. I am not sure what makes S-NBD2 "not suitable for a wide range of applications" since the reported properties of S-NBD2 seem to be pretty comparable to those from S-BTT in the current manuscript
3. Row 197-201: I am not sure if solvent exposure tests can be used to prove high degree of cross-linking since even linear polymers (such as polysulfide elastomers obtained from step-growth polymerization between inorganic polysulfides and organic dihalides) with high sulfur contents are often completely insoluble in organic solvents.
4. S-BTT prepared from a reaction mixture containing 70 wt% sulfur was compared to S-DVB also prepared from 70 wt% sulfur in the feed. Since the BTT monomer contains > 50 wt% sulfur while DVB contains no sulfur, the resulting polymers have significantly different sulfur contents. The difference in sulfur contents may be the cause of most of the improved properties reported in the current manuscript. More systematic studies comparing previous inverse vulcanization polymers with the

identical final sulfur contents is desirable.

Reviewer #2:

Remarks to the Author:

The manuscript by Lee et al reports on the synthesis of a novel sulfur derived hybrid polymer via inverse vulcanization and rigorous characterization of IR optical properties toward MWIR and LWIR imaging. The approach is novel and the authors should be credited for rigorous processing of free standing polymer windows to enable accurate optical evaluation of IR transmittance. The chemistry aspects toward improved IR transparency are interesting as the use of a tris-phenolic organic comonomer. The manuscript suffers from over-stated, in some cases, unsubstantiated claims in efforts to point out the significance of the work, which is unnecessary since the work is well done and of high quality. Addressing these points and of the minor but significant technical concerns below will warrant publication of this work in Nature Comm:

- 1) the RI values $n > 2.00$ are worrisome. The RI of sulfur maximizes at $n \sim 2.00$, so should not be exceeded when 20-wt% organic comonomer is present. RI characterization in this report are likely slightly inflated. If the authors really think the material is $RI > 2.0$, more rigorous characterization beyond ellipsometry are required (e.g., prism coupling, and focal length lens measurements.
- 2) the definition and calculation of corrected IR transmittance-transparency values are not rigorously included here and are likely oversimplified. This is non-trivial quantification what are essentially low transparency materials. Numerous places in the manuscript claim a %T but without validation of method and quantification, this is not valid. The revisions should include both computational and experimental validation, or remove these %T claims
- 3) NBD2 calculations from the SI should be removed unless experimentally validated
- 4) the term polymer lens is correctly used in the manuscript and SI and must be corrected. The authors made a polymer window, which is not a functioning free-form optic beyond simply transmission
- 5) the use of PMMA CO₂ laser designed LWIR imaging target should acknowledge the original work done by Kleine et al. AngewChem 2019, which the authors have implemented here, but not for the first time
- 6) NMR evidence of this polymer must be provided either as: (1) solid state ¹³C NMR spectroscopy, (2) reduced degraded thiphenolic fragments and characterization of the recovered organic units

REVIEWER COMMENTS

Reviewer #1

The authors describe polymers obtained from bulk polymerization of elemental sulfur and 1,3,5-benzenetrithiol and demonstrate that the materials have enhanced optical properties compared to previously reported sulfur-rich polymers obtained from inverse vulcanization. The synthesis described in the manuscript is simple and straightforward, and the resulting sulfur-rich polymers exhibit a combination of properties ($RI > 1.9$, $T_g \sim 100$ C, and some windows of transparency in MWIR and LWIR regions) that represent progress compared to IR-transparent polymeric materials reported previously.

Given that inverse vulcanization polymers such as S-DVB and S-NBD2 (both described in the references cited by the authors in the present manuscript) also exhibit high T_g , acceptable transparency in the MWIR and LWIR regions, and can be prepared without significant H₂S formation (the formation of H₂S during the synthesis of S-BTT reported in the present manuscript is mechanistically necessary and at least 3 eq of H₂S is released with respect to BTT), I am doubtful as to whether the S-BTT polymers represent significant advancement in the area. Formation of H₂S represents a formidable barrier towards scale-up and application of the materials for actual optical applications, and I am not sure if such risk is outweighed by an improvement of about 1.0 in the RI value.

One aspect of the current manuscript which would be of significant contribution to the field is in the comonomer design principle. The authors claim that the use of a symmetric comonomer improves the optical transparency of polymers in the IR region. The manuscript, however, does not contain control studies to show that this design principle is what causes the improved optical properties compared to S-DIB and S-DVB (which are also both obtained from symmetric comonomers). Also, many highly crosslinked polymers, including S-DVB, show additional IR absorptions attributed to overtones and/or structural confinement. The current manuscript, although hinting at the relationship between comonomer symmetry and the IR transparency of the polymer products, do not seem to provide much experimental and/or computational evidence to show that this indeed is a design principle that other researchers in the area can follow up on.

Reply to the Reviewer #1

We thank the reviewer 1 for the valuable commentary on how we could improve the manuscript. The S-DVB, which exhibits high T_g and MWIR transparency, and S-NBD2, which shows high T_g and LWIR transparency, are excellent inverse vulcanization polymers. We believe that our study is an original research that achieved high RI value and IR transmittance, particularly LWIR transmittance, while maintaining high T_g values by using a symmetric thiol cross-linker.

The ring-opening and radical formation of elemental sulfur requires high temperatures above the floor temperature (159 °C). Therefore, the copolymerization of elemental sulfur with various organic cross-linkers at low temperatures that can reduce H₂S production is a very important research topic. Recently, an inverse vulcanization in which the reaction temperature is lowered to 100-135 °C by introducing a catalytic pathway (X Wu et. al *Nature Communications* 2019) and a photoinduced inverse vulcanization at room temperature (J Jia et al. *Nature Chemistry* 2022) have been reported. Unfortunately, the reaction temperature of typical inverse vulcanization is still high, so toxic H₂S gas generation is unavoidable. We are aware of the significant amounts of H₂S by-products generated in inverse vulcanization with thiol cross-linker. Although H₂S by-products is inevitable at our reaction temperature, if we can successfully capture and convert these by-products, we expect it to be a further work in renewable energy (AG De Crisci et al. *International Journal of Hydrogen Energy* 2019 and M Dan et al. *Journal of Photochemistry & Photobiology C: Photochemistry Reviews* 2020).

We further prepared S-DVB and S-BTT polymers with almost equal sulfur percent (%). The S-DVB and S-BTT copolymers have been successfully prepared in window. However, the S-DIB polymers has low T_g and they were difficult to maintain the window shape in sulfur content of 90~70 wt%. Therefore, the comparison of optical properties described in the “Query 4” was centered around the S-DVB and S-BTT polymers. The results demonstrate that the use of a symmetric thiol cross-linker improves the IR transparency of the polymer. Please refer to the "Reply 4" below to detailed results. In addition, we have already described the DFT calculations using the model compounds in “Optical properties of the poly(S-*r*-BTT) copolymers” section of the manuscript and “G) DFT calculation of models compounds” section in supplementary information. The results indicate very low IR absorption due to the structural effect of S-BTT polymer.

The responses to some additional comments are listed below.

Query 1. Row 59-60 in the introduction reads “...their low refractive index usually make them

unsuitable for IR transmissive applications.”: I am not sure why $RI < 1.6$ makes a material unsuitable for IR applications as long as they show optical transparency in the IR region. Such medium- to low-index materials are excellent candidates for antireflective coatings for current inorganic IR lenses since the high intrinsic RI of inorganic materials are associated with their low transparency.

Reply 1: Thank you for your advice. We admit that the word “unsuitable” is inappropriate. As your comment, we revised the following on page 3, line 16-18 of the manuscript.

“However, their optical losses, due to IR absorption of C-H or heteroatom-hydrogen covalent bonds, and their low refractive index ($n < 1.6$), usually make them limited for various IR transmissive applications.^{6,7}”

Query 2. Row 89-90 in the introduction: More quantitative comparison between the previously reported S-NBD2 and S-BTT in the current manuscript would be desirable. I am not sure what makes S-NBD2 “not suitable for a wide range of applications” since the reported properties of S-NBD2 seem to be pretty comparable to those from S-BTT in the current manuscript.

Reply 2: The S-NBD2 and S-BTT polymers have a fundamental difference on how to approach the IR transparency. The S-NBD2 polymer excludes C=C bonds that induce strong absorption in the LWIR region. On the other hand, the S-BTT polymer exhibits simple IR absorption due to its symmetric structure despite the use of an aromatic cross-linker. We wanted to emphasize that the S-BTT with higher RI value and IR transmittance can be used for a wider range of applications.

As the reviewer mentioned, we admit that the word “not suitable” is inappropriate. So, we revised the following on page 5, line 1-2 of the manuscript.

“However, the refractive index (n) tended to be somewhat lower as the cross-linker content increased.”

Query 3. Row 197-201: I am not sure if solvent exposure tests can be used to prove high degree of cross-linking since even linear polymers (such as polysulfide elastomers obtained from step-growth polymerization between inorganic polysulfides and organic dihalides) with high sulfur contents are often completely insoluble in organic solvents.

Reply 3: Thank for the valuable comment. We agree with your comment. The results of the solubility test, referring to the previous study on highly cross-linked inverse vulcanization

polymers (S Park et al. *ACS Macro Letters* 2016, TS Kleine et al. *ACS Macro Letters* 2016 and DJ Parker et al. *Journal of Materials Chemistry A* 2017), were presented in the manuscript. In addition, we have already calculated the cross-linking density to further prove the high degree of cross-linking of the S-BTT polymer. The cross-linking density (ν_e) calculated from the storage modulus at the rubbery plateau was presented in Table S5 of supplementary information. The cross-linking density (ν_e) of the S-BTT polymer increased from 1294 to 2767 mol/m³ with increasing BTT content.

Table S5. Cross-linking density of poly(S-*r*-BTT) copolymers of varying sulfur content.

	S80-BTT20	S70-BTT30	S60-BTT40	S50-BTT50
ν_e^* (mol/m ³)	1294	1598	2488	2767

*Cross-linking density calculated as $\nu_e = E'/3RT$, where E' , R , and T are the storage modulus at rubbery plateau regime, ideal gas constant and temperature at E' , respectively.⁴ Each E' values were applied from DMA curves of Figure S23.

Query 4. S-BTT prepared from a reaction mixture containing 70 wt% sulfur was compared to S-DVB also prepared from 70 wt% sulfur in the feed. Since the BTT monomer contains > 50 wt% sulfur while DVB contains no sulfur, the resulting polymers have significantly different sulfur contents. The difference in sulfur contents may be the cause of most of the improved properties reported in the current manuscript. More systematic studies comparing previous inverse vulcanization polymers with the identical final sulfur contents is desirable.

Reply 4: Thank for the valuable advice. We performed additional experiments for systematic comparison with previous inverse vulcanization polymers with the identical final sulfur contents. We prepared S-DVB and S-BTT polymers with almost equal sulfur percent (%). The final sulfur percentage of S-DVB polymers with sulfur feed of 90~70 wt% were matched well with the S-BTT polymers with sulfur feed of 80~50 wt%. The elemental analysis results are summarized in Table S10 of supplementary information.

The refractive index and IR transmittance of the S-DVB and S-BTT windows were successfully measured. However, the S-DIB windows was difficult to measure due to their low T_g . Therefore, the comparison of optical properties according to the structure of the cross-linker was centered around DVB and BTT. We have added figures and tables related to this

results in the supplementary information. In addition, we have added the FT-IR spectra of S50-BTT50 window with the same sulfur percentage as S70-DIB30 and S70-DVB30 in Fig.3E&F of the manuscript.

Table S10. Elemental analysis of the S70-DIB30, poly(S-*r*-DVB) and poly(S-*r*-BTT) copolymers for varying sulfur content.

	Content (wt%)		Carbon (C) %	Hydrogen (H) %	Nitrogen (N) %	Sulfur (S) %	%
	Sulfur	Comonomer					
S70-DIB30	70	30	26.072	2.228	0.015	73.299	101.614
S90-DVB10	90	10	7.930	0.636	0.000	93.620	102.186
S85-DVB15	85	15	12.394	0.993	0.000	88.292	101.679
S80-DVB20	80	20	15.735	1.289	0.000	85.100	102.124
S70-DVB30	70	30	24.754	1.881	0.006	74.018	100.659
S80-BTT20	80	20	9.500	0.321	0.015	92.473	102.309
S70-BTT30	70	30	14.009	0.483	0.015	87.682	102.189
S60-BTT40	60	40	19.118	0.683	0.000	82.337	102.138
S50-BTT50	50	50	24.339	0.896	0.000	75.120	100.355

We measured the refractive index and IR transmittance of S-DVB windows with sulfur content 90-70 wt% and compared them to S-BTT windows with the identical final sulfur percentage. The results showed that the refractive index increased as the sulfur content increased for both S-DVB and S-BTT polymers (Table S8 and Figure S44). On the other hand, the IR transmittance was more affected by the structure of cross-linker than the final sulfur percentage. In Figure S46, the S-BTT windows shows much higher IR transmission (%) in both MWIR and LWIR than the S-DVB windows at the same final sulfur percentage. This difference is especially prominent in the LWIR region. Our claim that the use of a symmetric cross-linker improves the IR transparency of polymers is supported by these results.

Figure S24. Average refractive index curves of poly(S-r-DVB) films.

Table

S6.

	Wavelength (nm)	n_{TE}^a	n_{TM}^b	$n_{av.}^c$	Δn^d
S90-DVB10	637	1.955	1.955	1.955	0.000
	829	1.927	1.927	1.927	0.000
	1306	1.906	1.906	1.906	0.000
	1549	1.901	1.900	1.901	0.001
S85-DVB15	637	1.938	1.938	1.938	0.000
	829	1.910	1.910	1.910	0.000
	1306	1.888	1.888	1.888	0.001
	1549	1.884	1.883	1.884	0.000
S80-DVB20	637	1.919	1.918	1.919	0.001
	829	1.891	1.887	1.890	0.004
	1306	1.870	1.869	1.870	0.002
	1549	1.866	1.864	1.866	0.002
S70-DVB30	637	1.868	1.867	1.868	0.001
	829	1.843	1.843	1.843	0.001
	1306	1.824	1.822	1.823	0.002
	1549	1.820	1.818	1.819	0.001

Refractive indices of the poly(S-r-DVB) films at different wavelengths.

^aIn-plane refractive index, ^bOut-of-plane refractive index, ^cAverage refractive index calculated as $n_{av} = [(2n_{TE}^2 + n_{TM}^2)/3]^{1/2}$, ^dIn-plane/out-of-plane birefringence calculated as $\Delta n = n_{TE} - n_{TM}$.

Table S8. Optical properties of S70-DIB30, poly(S-*r*-DVB) and poly(S-*r*-BTT) films of varying sulfur content.

	Refractive indices and birefringence at 637 nm				V_{NIR}^e
	n_{TE}^a	n_{TM}^b	n_{av}^c	Δn^d	
S70-DIB30	1.861	1.861	1.861	0.000	19.8
S90-DVB10	1.955	1.955	1.955	0.000	18.9
S85-DVB15	1.938	1.938	1.938	0.000	18.1
S80-DVB20	1.919	1.918	1.919	0.001	18.2
S70-DVB30	1.868	1.867	1.868	0.001	19.0
S80-BTT20	2.001	2.001	2.001	0.000	16.5
S70-BTT30	1.991	1.988	1.990	0.002	16.3
S60-BTT40	1.968	1.967	1.968	0.001	17.2
S50-BTT50	1.945	1.942	1.944	0.003	16.8

^aIn-plane refractive index, ^bOut-of-plane refractive index, ^cAverage refractive index calculated as $n_{av} =$

$[(2n_{TE}^2 + n_{TM}^2)/3]^{1/2}$, ^dIn-plane/out-of-plane birefringence calculated as $\Delta n = n_{TE} - n_{TM}$. ^eAbbe's number is given by as $V_{NIR} = (n_{829} - 1)/(n_{637} - n_{1306})$.

Figure S44. Refractive indices by sulfur percent (%) of poly(*S-r*-DVB) and poly(*S-r*-BTT) films.

Figure S27. FT-IR transmission (%) spectra of the poly(*S-r*-DVB) windows (5 measurements) (a) S90-DVB10, (b) S85-DVB15, (c) S80-DVB20 and (d) S70-DVB30.

Figure S28. Average FT-IR transmission (%) spectra of poly(*S-r*-DVB) windows.

Figure S45. LWIR transmission (%) by sulfur percent (%) of the poly(*S-r*-DVB) and poly(*S-r*-BTT) windows.

Figure S46. FT-IR transmission (%) spectra of the poly(S-*r*-DVB) and poly(S-*r*-BTT) windows with same percentage of sulfur (a) S80-BTT20&S90-DVB10, (b) S70-BTT30&S85-DVB15, (c) S60-BTT40&S80-DVB20 and (d) S50-BTT50&S70-DVB30.

Reviewer #2

The manuscript by Lee et al reports on the synthesis of a novel sulfur derived hybrid polymer via inverse vulcanization and rigorous characterization of IR optical properties toward MWIR and LWIR imaging. The approach is novel and the authors should be credited for rigorous processing of free-standing polymer windows to enable accurate optical evaluation of IR transmittance. The chemistry aspects toward improved IR transparency are interesting as the use of a tris-phenolic organic comonomer. The manuscript suffers from over-stated, in some cases, unsubstantiated claims in efforts to point out the significance of the work, which is unnecessary since the work is well done and of high quality. Addressing these points and of the minor but significant technical concerns below will warrant publication of this work in Nature Comm:

Reply to the Reviewer #2

We greatly appreciate the comments of the reviewer 2. We tried to address the significant technical concerns you advised.

Query 1. the RI values $n > 2.00$ are worrisome. The RI of sulfur maximizes at $n \sim 2.00$, so should not be exceeded when 20-wt% organic comonomer is present. RI characterization in this report are likely slightly inflated. If the authors really think the material is $RI > 2.0$, more rigorous characterization beyond ellipsometry are required (e.g., prism coupling, and focal length lens measurements).

Reply 1: Thank for the valuable advice. According to our measurement results, the RI value of S80-BTT20 is 2.001 at 637 nm. We agree with your worrisome about our RI values and we do not think the materials is $RI > 2.0$. The S80-BTT20 had a sulfur content $> 92\%$ by elemental analysis. Thus, it can exhibit similar RI value of $n \sim 2.00$ similar to elemental sulfur. In addition, we measured the RI of bulk films using prism coupler at bulk thickness mode. The index of the bulk sample was calculated using the angular location of the knee in bulk mode (*Meticon.com*). The prism coupler's software automatically identifies the knee location on the intensity plot.

The experiment was repeated using the Prism coupler, the RI value of S80-BTT20 at 637 nm was confirmed to be close to 2.00. In addition to these assumptions, we have added recently published paper as references in manuscript (JH Hwang et al. *Advanced Optical Materials* 2023). The RI value in this paper also indicates 2.0 at 532 nm (90 wt% sulfur content).

Query 2. the definition and calculation of corrected IR transmittance-transparency values are not rigorously included here and are likely oversimplified. This is non-trivial quantification what are essentially low transparency materials. Numerous places in the manuscript claim a %T but without validation of method and quantification, this is not valid. The revisions should include both computational and experimental validation, or remove these %T claims.

Reply 2: Thank you for your advice. We acknowledge the "%T" expression that it is not necessarily the universal, or indeed the majority view. Therefore, we have refined the sentence in manuscript and numerous tables of supplementary information have been removed.

We removed the following on page 17 of the manuscript.

“We calculated the average IR transmission (T_{avg}) (%) of each of the mid-wave infrared (MWIR) and the long-wave infrared (LWIR) regions from the FT-IR spectra, and the results are summarized in Fig. 3C and Supplementary Table 19.”

We added the following new sentence on page 17, line 6-8 of the manuscript.

“The S-BTT windows exhibits numerous transparency peaks in the both MWIR and LWIR region for the overall sulfur content of 80~50 wt% (Fig 3C).”

We revised the following on page 17, line 8-12 of the manuscript.

“The S80-BTT20 window with the highest sulfur content showed the highest IR transmission (%) in both the MWIR and LWIR region. As the sulfur content decreased, the IR transmittance of the S-BTT windows decreased. However, the IR transmittance of the S50-BTT50 window with the lowest sulfur content was significantly higher than the S70-DIB30 and S70-DVB30 windows (Fig. 3E and 3F).”

We revised the following on page 22, line 13-16 of the manuscript.

“These findings confirmed the S90-BTT10 window exhibited unprecedentedly high IR transmittance in both the MWIR and LWIR region (Supplementary Figure 13), but an IR imaging experiment was not considered because of the presence of unreacted sulfur.”

We revised the following on page 23, line 3-6 of the manuscript.

“The transparent plastic PMMA failed to visualize the USAF target due to its low transmittance, while the S70-DIB30 and S70-DVB30 windows were capable of imaging in the MWIR region, with results consistent with previous studies.”

We revised the following on page 23, line 8-10 of the manuscript.

“The S80-BTT20 window with the highest transmission (%) in MWIR region provided the clearest image, and the S50-BTT50 window with the lowest sulfur content (50 wt%) also provided a clear image (**Fig. 4F**).”

We revised the following on page 26, line 5-7 of the manuscript.

“As with the MWIR imaging experiments, a decrease in transmittance of LWIR was observed as the sulfur contents of the poly(S-*r*-BTT) copolymers decreased, however, high-quality LWIR imaging was achieved for overall sulfur contents of 80~50 wt%.”

We revised the following on page 26, line 11-12 of the manuscript.

“The S50-BTT50 window with the lowest sulfur content (50 wt%) was also sufficiently transparent in the LWIR region (**Fig. 4G**).”

Query 3. NBD2 calculations from the SI should be removed unless experimentally validated.

Reply 3: As your comment, we have removed calculations for NBD2, TIB, TVSn that have not been experimentally validated from the supplementary information.

Query 4. the term polymer lens is correctly used in the manuscript and SI and must be corrected. The authors made a polymer window, which is not a functioning free-form optic beyond simply transmission.

Reply 4: According to the comment of the reviewer, we have corrected use of the term “polymer lens” to “polymer window” in manuscript and supplementary information.

Query 5. the use of PMMA CO₂ laser designed LWIR imaging target should acknowledge the original work done by Kleine et al. *Angew Chem* 2019, which the authors have implemented here, but not for the first time.

Reply 5: We indicated the original work of the LWIR imaging target in the revised manuscript and supplementary information (Kleine et al. *Angewandte Chemie International Edition* 2019). We revised the following on page 26 (&28), line 1 (&10) of the manuscript.

“For LWIR imaging experiments with various patterned PMMA masks¹⁷, an FLIR camera with the transmission range of 7.5-13 μm was used.” and “Various patterned PMMA masks were prepared as target for LWIR imaging.¹⁷”

We also revised the following on page 4, line 10 of the supplementary information.

“Various patterned PMMA masks were prepared as target for LWIR imaging.¹⁷”

Query 6. NMR evidence of this polymer must be provided either as: (1) solid state ^{13}C NMR spectroscopy, (2) reduced degraded thiophenolic fragments and characterization of the recovered organic units.

Reply 6: We thank for the valuable comment. We further performed solid state ^{13}C NMR spectroscopy with NMR evidence of S-BTT polymers (70~50 wt% sulfur content). The results have included in the supplementary information.

Figure S14. Solid-state ^{13}C NMR CP/MAS spectra of poly(S-r-BTT) copolymers.

Two signals corresponding to aromatic carbons were observed in the spectrum of the poly(*S-r*-BTT) copolymers. As shown in Figure S14, the peak at ca. 140 ppm and broad peaks at 120-130 ppm were assigned to C_{ph} (Co) and C_{ph-H} (Co), respectively. The peaks related to other organic moieties were not observed.

Reviewers' Comments:

Reviewer #1:

Remarks to the Author:

I have reviewed the revised manuscript. There are a few detailed points that I still find to be of slight concern:

1. The values of n nearing 2.0 are actually higher than the refractive index of pure molten sulfur (~ 1.86 to 1.94 depending on wavelength). A rigorous explanation for why BTT-sulfur copolymers have higher RI than pure sulfur need to be provided. One possibility may be that the density of the polymer is enhanced relative to molten sulfur due to cross-linking by BTT, which could be resolved from density measurements and evaluating the effect of density on refractive index through Lorentz-Lorenz relation.

2. The birefringence of 0 is extraordinary and have never been observed in sulfur-derived polymers. Since birefringence is usually reported to four decimal places, is this due to rounding the numbers?

3. More details of prism coupler measurements (light source, wavelength, type of prism used, temperature, etc).

4. Do the wavelength-dependent refractive index data fit the Cauchy equation? If so, Cauchy coefficients (n at infinite wavelength and dispersion coefficient) would be beneficial for comparing the optical properties of S-BTT polymers to other previously reported high index IR polymers.

5. The authors suggest that the polymerization proceeds through a radical mechanism wherein thermally generated sulfur radicals first remove hydrogen from -SH groups in BTT. However, polar mechanisms are also possible under these conditions, where the ring-opening of elemental sulfur occurs through nucleophilic attack by the -SH groups. Such polar mechanisms have been suggested in some of the early work leading to inverse vulcanization (*Angew. Chem. Int. Ed.* 2011, 50, 11409 – 11412). Since the work by the authors is distinct from previous inverse vulcanization polymers that involve addition of sulfur radicals to alkenes and alkynes, providing the mechanism of polymerization would provide a useful insight for expanding the comonomer scope in sulfur copolymers.

6. Reaction of sulfur and BTT is carried out at 185 degrees, which is a temperature at which S80-BTT20 undergoes thermal degradation as shown in the TGA data. Yet, no particular difference in sulfur content trends is shown in the elemental analysis data.

Reviewer #2:

Remarks to the Author:

The revised manuscript is suitable for publication. In response to reviewer #1's review of the manuscript, I would deem this perspective likely arises from this referee's limited experience in polymer optics and IR polymers.

REVIEWER COMMENTS

Reply to the Reviewer 1

We greatly appreciate your comments. We tried to address the slight concern you advised. The responses are listed below.

1. The values of n nearing 2.0 are actually higher than the refractive index of pure molten sulfur (~ 1.86 to 1.94 depending on wavelength). A rigorous explanation for why BTT-sulfur copolymers have higher RI than pure sulfur need to be provided. One possibility may be that the density of the polymer is enhanced relative to molten sulfur due to cross-linking by BTT, which could be resolved from density measurements and evaluating the effect of density on refractive index through Lorentz-Lorenz relation.

Reply 1: Thank for the valuable advice. The refractive index (RI) of sulfur maximizes at $n \sim 2.0$. According to our measurement results using prism coupler, the RI value of S80-BTT20 is 2.0013 at 637 nm. We agree with your advice about our RI values and we do not think the materials is $RI > 2.0$. Elemental analysis of S80-BTT20 showed a very high sulfur content (92.473 %). Thus, it can exhibit similar RI value of sulfur. We measured the density of the S-BTT polymers as your comment. According to the results, the RI values of S-BTT polymers increases as the density increases. And the density of S80-BTT20 was higher than liquid sulfur (1.819 g/cm^3) (Tassi F et al. *Poás Volcano: The Pulsing Heart of Central America Volcanic Zone* 2019) These results accord closely with Lorentz–Lorenz equation.

$$\frac{n^2 - 1}{n^2 + 2} = \frac{R}{M} \rho$$

where ρ is the density, n is the refractive index, R is the molar refraction, M is the molecular weight.

	Content (wt%)		Density (g/cm^3)*	n_{av} at 637 nm
	Sulfur	BTT		
S80-BTT20	80	20	1.827	2.001
S70-BTT30	70	30	1.797	1.990
S60-BTT40	60	40	1.758	1.968
S50-BTT50	50	50	1.709	1.944

* Average of 5 measurements, using density accessory kit for analytical balance (Mettler Toledo, U.S.A)

2. The birefringence of 0 is extraordinary and have never been observed in sulfur-derived polymers. Since birefringence is usually reported to four decimal places, is this due to rounding the numbers?

Reply 2: We revised the Table of RI value to 3 decimal places in first revision. As your comment, birefringence is usually given to 4 decimal places, so we revised the Table of RI value of supporting information as follows with raw data.

Table S6. Refractive indices of the poly(*S-r*-DVB) films at different wavelengths.

	Wavelength (nm)	n_{TE}^a	n_{TM}^b	$n_{av.}^c$	Δn^d
S90-DVB10	637	1.95502	1.95471	1.95492	0.00031
	829	1.92686	1.92658	1.92677	0.00028
	1306	1.90602	1.90578	1.90594	0.00024
	1549	1.90114	1.90024	1.90084	0.00090
S85-DVB15	637	1.93835	1.93816	1.93829	0.00019
	829	1.90991	1.90982	1.90988	0.00009
	1306	1.88835	1.88777	1.88816	0.00058
	1549	1.88362	1.88348	1.88357	0.00014
S80-DVB20	637	1.91888	1.91807	1.91861	0.00081
	829	1.89092	1.88673	1.88952	0.00419
	1306	1.87033	1.86857	1.86974	0.00176
	1549	1.86632	1.86391	1.86552	0.00241
S70-DVB30	637	1.86818	1.86696	1.86777	0.00122
	829	1.84342	1.84287	1.84324	0.00055
	1306	1.82403	1.82198	1.82335	0.00205
	1549	1.81959	1.81844	1.81921	0.00115

^aIn-plane refractive index, ^bOut-of-plane refractive index, ^cAverage refractive index calculated as $n_{av} = [(2n_{TE}^2 + n_{TM}^2)/3]^{1/2}$, ^dIn-plane/out-of-plane birefringence calculated as $\Delta n = n_{TE} - n_{TM}$.

Table S7. Refractive indices of the poly(S-*r*-BTT) films at different wavelengths.

	Wavelength (nm)	n_{TE}^a	n_{TM}^b	$n_{av.}^c$	Δn^d
S80-BTT20	637	2.00130	2.00105	2.00122	0.00025
	829	1.96723	1.96710	1.96719	0.00013
	1306	1.94278	1.94254	1.94270	0.00024
	1549	1.93784	1.93564	1.93711	0.00220
S70-BTT30	637	1.99078	1.98829	1.98995	0.00249
	829	1.95721	1.95468	1.95637	0.00253
	1306	1.93199	1.93071	1.93156	0.00128
	1549	1.92874	1.92595	1.92781	0.00279
S60-BTT40	637	1.96797	1.96686	1.96760	0.00111
	829	1.93816	1.93537	1.93723	0.00279
	1306	1.91345	1.91175	1.91288	0.00170
	1549	1.90919	1.90634	1.90824	0.00285
S50-BTT50	637	1.94536	1.94207	1.94426	0.00329
	829	1.91428	1.91231	1.91362	0.00197
	1306	1.89107	1.88900	1.89038	0.00207
	1549	1.88680	1.88591	1.88650	0.00089

^aIn-plane refractive index, ^bOut-of-plane refractive index, ^cAverage refractive index calculated as $n_{av} = [(2n_{TE}^2 + n_{TM}^2)/3]^{1/2}$, ^dIn-plane/out-of-plane birefringence calculated as $\Delta n = n_{TE} - n_{TM}$.

3. More details of prism coupler measurements (light source, wavelength, type of prism used, temperature, etc).

Reply 3: Refractive indices of polymer films were measured using a prism coupler (PC-2000, Metricon Corporation, U.S.A). All measurements were performed under laboratory condition at room temperature.

- Light source: He-Ne laser light source
- Operating wavelength: 636.5 nm, 828.7 nm, 1305.8 nm, 1549.5 nm
- Prism type: 200-P-2
- Index range: 1.70~2.45

The index of the bulk material was calculated using the angular location of the knee in bulk mode. The prism coupler's software automatically identifies the knee location on the intensity plot (*Metricon.com*).

4. Do the wavelength-dependent refractive index data fit the Cauchy equation? If so, Cauchy coefficients (n at infinite wavelength and dispersion coefficient) would be beneficial for comparing the optical properties of S-BTT polymers to other previously reported high index IR polymers.

Reply 4: The wavelength dependence of average refractive indices was fitted to the simplified Cauchy's formula. The results are shown in Fig.3A of the manuscript. We agree with your comment that the Cauchy coefficients would be beneficial for comparing the optical properties

of S-BTT polymers with other high index IR polymers. So, the Cauchy coefficients (n at infinite wavelength and dispersion coefficient) were added to RI table in the supplementary information.

Table S8. Optical properties of S70-DIB30, poly(S-*r*-DVB) and poly(S-*r*-BTT) films of varying sulfur content.

	Refractive indices and birefringence at 637 nm				V_{NIR}^e	n_∞^f	D^f [$\times 10^4$]
	n_{TE}^a	n_{TM}^b	n_{av}^c	Δn^d			
S70-DIB30	1.86103	1.86100	1.86102	0.00003	19.8	1.8053	2.2459
S90-DVB10	1.95502	1.95471	1.95492	0.00031	18.9	1.8899	2.6131
S85-DVB15	1.93835	1.93816	1.93829	0.00019	18.1	1.8723	2.6574
S80-DVB20	1.91888	1.91807	1.91861	0.00081	18.2	1.8542	2.5732
S70-DVB30	1.86818	1.86696	1.86777	0.00122	19.0	1.8093	2.3661
S80-BTT20	2.00130	2.00105	2.00122	0.00025	16.5	1.9239	3.1034
S70-BTT30	1.99078	1.98829	1.98995	0.00249	16.3	1.9140	3.0436
S60-BTT40	1.96797	1.96686	1.96760	0.00111	17.2	1.8959	2.8961
S50-BTT50	1.94536	1.94207	1.94426	0.00329	16.8	1.8740	2.8243

^aIn-plane refractive index, ^bOut-of-plane refractive index, ^cAverage refractive index calculated as $n_{\text{av}} = [(2n_{\text{TE}}^2 + n_{\text{TM}}^2)/3]^{1/2}$, ^dIn-plane/out-of-plane birefringence calculated as $\Delta n = n_{\text{TE}} - n_{\text{TM}}$, ^eAbbe's number is given by as $V_{\text{NIR}} = (n_{829} - 1)/(n_{637} - n_{1306})$, ^fCalculated refractive index at infinite wavelength determined by fitting with the simplified Cauchy's formula ($n_\lambda = n_\infty + D/\lambda^2$).

- The authors suggest that the polymerization proceeds through a radical mechanism wherein thermally generated sulfur radicals first remove hydrogen from -SH groups in BTT. However, polar mechanisms are also possible under these conditions, where the ring-opening of elemental sulfur occurs through nucleophilic attack by the -SH groups. Such polar mechanisms have been suggested in some of the early work leading to inverse vulcanization (Angew. Chem. Int. Ed.2011,50, 11409–11412). Since the work by the authors is distinct from previous inverse vulcanization polymers that involve addition of sulfur radicals to alkenes and alkynes, providing the mechanism of polymerization would provide a useful insight for expanding the comonomer scope in sulfur copolymers.

Reply 5: Thank for the valuable comment. We referred to previous studies on the interactions of sulfur with thiol-based compounds in the manuscript. Thiol compounds are commonly used for the synthesis of polymeric polysulfides (NP Tarasova et al. *RSC Adv* 2021).

Scheme 5 Dithiols used in the reaction with elemental sulphur.

These studies synthesized polymer solutions via a condensation reaction between the thiol monomer and sulfur in a toluene/carbon disulfide mixture (A Bhargav et al. *ACS Appl. Mater. Interfaces* 2018 and W Guo et al. *Nat. Commun* 2021). In our study, molten sulfur acted as a comonomer and reaction solvent. 1,3,5-benzenetrithiol (BTT) has high reactivity of three thiol groups, facilitated by the conjugated nature of the benzyl ring, favor the elimination of protons on a condensation reaction with elemental sulfur (A Bhargav et al. *ACS Appl. Energy Mater* 2018 and Y Zhang et al. *Polym. Chem* 2019). We are conducting further studies on thiol cross-linkers, which are distinct from previous inverse vulcanization polymers using alkene and alkyne cross-linkers. Sooner or later we will provide a detailed mechanism through further studies. Like you comment, we hope that our further studies will provide useful insights for expanding the comonomer scope in inverse vulcanization.

6. Reaction of sulfur and BTT is carried out at 185 degrees, which is a temperature at which S80-BTT20 undergoes thermal degradation as shown in the TGA data. Yet, no particular difference in sulfur content trends is shown in the elemental analysis data.

Reply 6: As you mentioned, the TGA results show that the S80-BTT20 undergoes thermal degradation beginning below 185 °C and with a total mass loss of 99 %. Most high sulfur content polymers decrease thermal stability as the cross-linker content decreases. The S80-BTT20 also lack of thermal stability with a low T_g (<room temperature). However, the degradation onset temperature of elemental sulfur is above 185 °C, and the reaction of sulfur and BTT carried well. The Excessive processing time at high temperature can deteriorate the thermodynamic properties of sulfur polymers. So, although the inverse vulcanization was carried out at 185 °C, we synthesized the S-BTT with a short reaction time in an hour.

In addition, we present the results of elemental analysis of three measurements of S80-BTT20. The results of all measurements are closely similar.

	Content (wt%)		Carbon (C) %	Hydrogen (H) %	Nitrogen (N) %	Sulfur (S) %	%
	Sulfur	Comonomer					
			9.182	0.308	0.000	92.831	102.321
S80-BTT20	80	20	9.500	0.321	0.015	92.473	102.309
			8.850	0.308	0.000	93.249	102.407

REVIEWER COMMENTS

Reply to the Reviewer 2

We are most grateful to you for your comments that our revised manuscript is suitable for publication. Your advice was very helpful.

Reviewers' Comments:

Reviewer #1:

Remarks to the Author:

I have reviewed the revised manuscript and now feel that the manuscript is suitable for publication. I would like to thank the authors for detailed and careful investigations that provide, in my opinion, new insights in this area of research.